# Facile Preparation of Pd/UiO-66-v for the Conversion of Furfuryl Alcohol to Tetrahydrofurfuryl Alcohol under Mild Conditions in Water

**DOI:** 10.3390/nano9121698

**Published:** 2019-11-28

**Authors:** Yanliang Yang, Dongsheng Deng, Dong Sui, Yanfu Xie, Dongmi Li, Ying Duan

**Affiliations:** 1Henan Key Laboratory of Function-Oriented Porous Material, College of Chemistry and Chemical Engineering, Luoyang Normal University, Luoyang 471934, China; 2College of Food and Drug, Luoyang Normal University, Luoyang 471934, China

**Keywords:** furfuryl alcohol, hydrogenation, tetrahydrofurfuryl alcohol, Pd/UiO-66-v, water solvent

## Abstract

The hydrogenation of furan ring in the biomass-derived furans is of great importance for the conversion of biomass to valuable chemicals. Fabrication of high activity and selectivity catalyst for this hydrogenation under mild conditions was one of the focuses of this research. In this manuscript, UiO-66-v, in which vinyl bonded to the benzene ring, was first prepared. Then, the uniformly distributed vinyl was used as the reductant for the preparation of Pd/UiO-66-v. The catalyst was characterized by X-ray diffraction, thermogravimetric, N_2_ physical adsorption/desorption, X-ray photoelectron spectroscopy, scanning electron microscope, transmission electron microscopy, energy dispersive spectrometer elemental mappings, and inductively coupled plasma atomic emission spectroscopy to find the Pd/UiO-66-v had a narrow palladium nanoparticles size of 3–5 nm and maintained the structure and thermal stability of UiO-66-v. The Pd/UiO-66-v was used for the hydrogenation of furfuryl alcohol to tetrahydrofurfuryl alcohol in water. 99% conversion of furfuryl alcohol was obtained with 90% selectivity to tetrahydrofurfuryl alcohol after reacted at 0.5 MPa H_2_, 303 K for 12 h. The Pd/UiO-66-v was proved to be effective for the hydrogenation of furan ring in furans and could be used for at least five times.

## 1. Introduction

Furfural, a renewable compound from dehydration of pentoses, is produced in millions of tons scale in industry every year. Furfural was called gold from garbage, as most of the raw material for the production of furfural was agricultural waste such as corncobs, cotton hulls, sugar cane bagasse, and more [1]. The conversion of furfural to valuable chemicals was attracting more attention as the consuming in fossil resources [2,3,4,5,6,7]. The main products from furfural were cyclopentanone [8,9,10,11,12], furfuryl alcohol (FA) [13,14,15,16], tetrahydrofurfuryl alcohol (THFA) [17,18,19,20,21,22,23,24,25,26,27,28,29,30], 1,2-pentanediol [31,32,33,34,35], furan [36,37], and methyl furan [38]. The THFA was a typical product from furfural through the hydrogenation of C=C and C=O in furfural. The THFA is a transparent low toxic liquid of high boiling point and is miscible with water. The THFA could be a good candidate for green solvent due to the high chemical and thermal stability as well as biologic degradability. The THFA is also an intermediate for high value-added compounds such as pyridine [39], dihydropyran [40], and tetrahydrofuran [41]. Additionally, it was reported that 1,5-pentanediol, a promising monomer for the plastics industry, could be obtained from the hydrogenolysis of THFA [42,43,44,45,46].

There was plenty of literature dealing with the conversion of furfural to THFA directly or via FA as an intermedia in two steps. Tomishige et al. gave a report on the direct conversion of furfural to THFA on Ni/SiO_2_ with Ni particle size <4 nm. The yield of THFA could reach 94% [19]. They also found that the reaction could be conducted in the aqueous phase catalyzed by Pd-Ir/SiO_2_ [18]. Guan et al. achieved the total hydrogenation of furfural to THFA by the physical mix of Pd/Al_2_O_3_ and Ru/ZrO_2_ [20]. The reaction could be conducted under mild conditions and was believed to proceed with tetrahydrofurfural as intermediate. FA was the main product from the selective hydrogenation of furfural with mature technology in industrial. The conversion of FA to THFA suffered the scientific problem of the hydrogenation of the furan ring, while both the aldehyde and furan ring should be hydrogenated in the process for the conversion of furfural to THFA directly. As a result, the conversion of FA to THFA was of great importance both in industrial and theoretical. More moderate conditions may be needed in the conversion of FA to THFA. Wang et al. performed this conversion using NiCo bimetallic alloy as catalysts. The synergistic effect in the CoNi alloy was responded for the high performance, and 99% yield of THFA was obtained at 353 K, 3 MPa H_2_ in ethanol [25]. Guan et al. got 98% conversion of FA and 98% selectivity to tetrahydrofurfuryl alcohol under mild condition in ethanol [47]. In a survey of literature, the yield of THFA could reach a high value in both the direct hydrogenation of furfural procedure and the two steps procedure via FA as an intermedia [17,18,19,20,21,22,23,24,25,26,27,28,29,30]. The weakening of the dependence on harsh reaction conditions (high temperature and pressure, organic solvent) was one of the interest points for the production of THFA.

Due to the high specific surface area, three-dimensional uniform pore structure and easy to be functionalized, the metal organic frameworks (MOFs) had emerged as an excellent candidate for catalyst or support [48,49,50,51,52,53]. The combination of metal nanoparticles and MOFs often gave the production of high-performance catalysts. Liang et al. prepared Pd@MIL-101(Cr)-NH_2_ for the hydrogenation of furfural in water [21]. Near 100% yield of THFA was obtained at 313 K, 2 MPa H_2_. Hensen et al. obtained Ru/UiO-66 through the reduction of RuCl_3_ by N_2_H_4_·H_2_O [14]. The Ru/UiO-66 showed high activity in the hydrogenation of furfural to FA. During the preparation of MOFs supported metal nanoparticles, the mild conditions, low temperature, and near-neutral pH for the reduction of metal precursors was a favor to avoid the potential damage and impact on structures of MOFs. In our previous work, the alkenyl was proven to be an ideal reductant for PdCl_2_ [54,55,56,57]. The reduction reaction could conduct at room temperature in neutral water. Herein, we gave a report on the preparation of Zr-MOF with UiO-66 topology by the condensation of 2-vinylterephthalic acid and zirconium tetrachloride. As the uniform distribution of vinyl in the synthesized UiO-66-v, uniform dispersion of Pd nanoparticles were acquired by just stirring the mixture of PdCl_2_ solution and as-synthesized UiO-66-v for 12 h. The prepared Pd/UiO-66-v exhibited high activity for the conversion of FA to THFA at 303 K, 0.5 MPa H_2_ in water.

## 2. Materials and Methods 

### 2.1. Materials

FA was obtained from Sinopharm Chemical Reagent Co., Ltd., Shanghai, China. Before being used, the FA was purified by vacuum distillation. PdCl_2_, ZrCl_4_, 2-methylfuran and 2,5-dimethylfuran was bought from J&K Chemical Ltd. Beijing, China, 2-vinylterephthalic acid was obtained from Zhengzhou alpha Chemical Co., Ltd. Zhengzhou, China, N, N-Dimethylformamide (DMF), ethanol, NaBH_4_, 5-methyl furfural, and 5-hydroxymethylfurfural was purchased from Aladdin Chemistry Co. Ltd. Shanghai, China. The 5-methylfuran-2-methanol and 2,5-di(hydroxymethyl)furan were prepared by the reduction of 5-methyl furfural and 5-hydroxymethylfurfural by NaBH_4_. Unless otherwise specified, the reagent was used as received. 

### 2.2. Preparation of Catalysts

UiO-66-v was prepared according to the previous report [58]. Typically, ZrCl_4_ was first dissolved in a mixture solvent of DMF and HCl (V_DMF_:V_HCl_ = 5:1, 15 mL) and then 2-vinylterephthalic acid (2.0 mmol) dissolved in DMF (25 mL) was added into the mixture and sonicated for 20 min. After that, the mixture was transferred into a 50 mL Teflon-lined steel autoclave. The autoclave was heated to 353 K and kept at 353 K for 12 h and then cooled to room temperature naturally. The white mixture was washed first by DMF (50 mL × 2) and then ethanol (50 mL × 3). The UiO-66 was obtained as a white powder after the precipitate was dried under a vacuum for 12 h at 363 K.

Pd/UiO-66-v was prepared through the reduction of PdCl_2_ by vinyl at mild conditions. Typically, UiO-66-v (0.20 g) was added to water (50 mL) and stirred for 30 min at 353 K. Then, dilute hydrochloric acid aqueous solution of PdCl_2_ (Pd: 0.5 wt.%, 1.00 g) was added to the mixture and stirred for another 12 h at 353 K to afford black precipitate. The precipitate was first washed with water (50 mL × 3) and then ethanol (50 mL × 3). Finally, it was dried under a vacuum for 12 h at 363 K.

### 2.3. Characterization of Catalysts

The powder X-ray diffraction (XRD) patterns were obtained on a Rigaku D/Max 2500/PC powder diffractometer (Tokyo, Japan) at a scanning rate of 5°·min^−1^ using Cu Kα radiation (λ = 0.15418 nm) at 40 kV and 40 mA. The thermogravimetric (TG) analysis was conducted on PerkinElmer Diamond TG/DTA6300 (Waltham, MA, USA). The temperature range was 323 K to 1073 K with the heating rate was 5 K·min^−1^. The N_2_ physical adsorption/desorption curves were collected on a Micromeritics ASAP 2040 apparatus (Narcross, GA, USA). Before measurement, the samples (50.0 mg) were pretreated at 423 K. The X-ray photoelectron spectroscopy (XPS) spectra were collected on a Thermo Fisher K-alpha (Waltham, MA, USA) with an Al Ka (1486.6 eV) radiation source. The pass energy was 200.0 eV for survey scan while 50 eV for high-resolution spectrum. The peak areas were constraint as 3d_5/2_:3d_3/2_ = 3:2 and 2p_3/2_:2p_1/2_ = 2:1. The 80% Lorentzian-Gaussian was selected as the shape line. The scanning electron microscope (SEM) images were taken on FEI Nova Nano-SEM (Hillsboro, OR, USA) with 20 kV HD. Transmission electron microscopy (TEM) images were taken by FEI Tecnai G2 F20 microscope (Hillsboro, OR, USA) with 200 kV accelerating voltage. The high-resolution transmission electron microscope (HRTEM) and energy dispersive spectrometer (EDS) elemental mappings were taken by JEM-2100F (Tokyo, Japan) equipped with an Oxford x-met8000 detector. Before test, the sample was ultrasonic in ethanol. The content of Pd was determined by inductively coupled plasma atomic emission spectroscopy (ICP-AES) by Agilent 7700 (Santa Clara, CA, USA).

### 2.4. Catalytic Reactions

The catalytic reaction was taken in a 20 mL stainless-steel autoclave with a glass lining. In a typical procedure, FA aqueous solution (1 mmol, 2 mL), Pd/UiO-66-v (10.0 mg) and magneton was added into the autoclave. After sealed and purged with H_2_ for four times to exclude the air, the autoclave was inflated with H_2_ to the desired pressure. The autoclave was then put in a magnetic stirring oil bath at 303 K for a certain time. After the reaction, internal standard (decane) was added into the mixture and then the mixture was diluted by ethanol. The liquid phase was collected for analysis after centrifugation. The turnover frequency (TOF) was calculated based on all the Pd atoms in the catalysts and the whole reaction time.

### 2.5. Product Analysis

The qualitative analysis was conducted by GC on a Shimadzu GC-2014 (Tokyo, Japan) equipped with a SH-Rtx-5 column (30 m × 0.32 mm × 0.25 μm). The column oven was first kept at 333 K for 4 min and then increased to 523 K at a rate of 20 K·min^−1^ and kept at 523 K for 1.5 min. Decane was used as the internal standard. The qualitative analysis was performed on Shimadzu GC/MS-TQ8040 (Tokyo, Japan) equipped with an InerCap 17MS column (30 m × 0.25 mm × 0.25μm). The column oven was first kept at 323 K for 1 min, and then increased to 473 K at a rate of 40 K·min^−1^ and increased to 553 K at a rate of 15 K·min^−1^ and kept at 553 K for five min. The obtained mass spectrum was shown in Appendix A (Appendix A). 

## 3. Results and Discussion

### 3.1. Catalyst Characterization

The Zr-MOF was prepared according to the previous reported procedure with modification that the 2-vinylterephthalic acid was used instead of terephthalic acid for the preparation of UiO-66-v. The XRD pattern of UiO-66-v showed that the as-synthetized Zr-MOF have the topology of UiO-66 comparing to the XRD pattern of simulated UiO-66 (Figure 1). The Pd was introduced to the UiO-66-v by the reaction of PdCl_2_ and vinyl in the UiO-66-v in water. The reduction procedure could be conducted just by stirring the UiO-66-v in the aqueous solution of PdCl_2_. As no harsh conditions such as high temperature or base was needed, the resulted Pd/UiO-66-v maintained the frame structure of UiO-66-v. No difference could be observed between the XRD patterns of UiO-66-v and Pd/UiO-66-v (Figure 1). Additionally, no crystal plane diffraction peak corresponding to metallic Pd was founded in the XRD pattern of Pd/UiO-66-v. To find out why there was no signal for Pd in the XRD pattern, the HRTEM picture of Pd/UiO-66-v was taken (Appendix A). It was found that not all the Pd nanoparticles had good crystallinity. Part of the Pd nanoparticles had amorphous state. On the other hand, the ICP-AES showed the Pd content was 2.42 wt.% (Table 1, Entry 2). Based on these, the absence of Pd signal in the XRD pattern should be caused by the low content of Pd, the amorphous state for part of the Pd nanoparticles and the small particle size of Pd. The thermal stability of UiO-66-v and Pd/UiO-66-v was studied by the thermogravimetric analysis, and the results were shown in Appendix A. There had a little weight loss at 373 K in both samples, which should be caused by the residual solvent in the samples during the preparation process. Then, the two samples had sharp weight loss when the temperature was close to 773 K. This was caused by the decomposition of UiO-66-v at this temperature. The results showed the high thermal stability for the framework of UiO-66-v and the introduction of Pd did not affect the stability of UiO-66-v.

The BET surface area and pore structure of as-synthesized UiO-66-v and Pd/UiO-66-v were characterized by N_2_ adsorption/desorption isotherms as shown in Figure 2. The BET surface was 862 m^2^·g^−1^ for UiO-66-v, which was closed to that for the UiO-66 modified with a single group (-NH_2_, -NO_2_ and -OH) reported in the literature (Table 1, Entry 1) [58,59,60]. The micropore area was calculated by the t-plot method and the area was 726 m^2^·g^−1^, indicating the surface area was most contributed by the micropore area. After the introduction of Pd, both the BET surface area and the micropore area calculated by the t-plot method decreased slightly and was accompanied by the decrease in pore volume. The reason should be the in presence of partly Pd in the pore of UiO-66. The practice Pd content was determined by ICP-AES, and the value (2.42 wt.%) was very close to the theoretical value (Table 1, Entry 2). After reaction, the Pd content was maintained showed the catalyst was stable under the mild reaction conditions (2.35 wt.%). 

The morphology of samples was characterized by the electron microscope. The UiO-66-v and Pd/UiO-66-v both crystallized nanocrystals with a size distribution of around 80 nm revealed by the scanning electron microscope (SEM) image (Appendix A). The introduction of Pd had no influence on the morphology of UiO-66-v based on the SEM results. The Pd nanoparticles had a narrow size distribution between 3–5 nm by counting the diameters of about 250 Pd nanoparticles in several transmission electron microscope (TEM) images (Figure 3, Appendix A). The narrow size distribution of Pd should be caused by the uniform distribution of the reductant. The EDS elemental mappings for Pd/UiO-66-v also showed that the Pd homogenously distributed in the sample (Figure 3e, Appendix A). The vinyl, used as the reductant, was linked by the carbon-carbon bond to the benzene ring in the structure of UiO-66-v. The number of metallic Pd atoms is limited in a given region caused by the limited number of vinyl for the reduction reaction of PdCl_2_. The used catalyst was characterized by the TEM. The Pd nanoparticles maintained the narrow size distribution, but enlarged slightly after the reaction (Figure 3c,d).

The surface chemical composition of Pd/UiO-66-v was characterized by X-ray photoelectron spectroscopy (XPS). The survey scan showed the surface was composed of Pd, Zr, C, and O (Appendix A). There was partial overlap in the high-resolution spectrum of Pd 3d and Zr 3p (Figure 4a, Table 2). After the curve fitting process, two binding energy peaks cantered at 335.4 eV and 340.6 eV were observed corresponding to the 3d_5/2_ and 3d_3/2_ of Pd(0), respectively [59,60,61]. The low doublet separation for Pd (5.2 eV) should be caused by the interaction between Pd and Zr. Obviously, the PdCl_2_ was reduced to Pd(0) successfully by the vinyl. The additional two peaks at 333.5 eV and 347.0 eV could be assigned to the Zr 3p_3/2_ and 3p_1/2_, respectively [59,60]. The presence of Zr was also proved by the binding energy peaks at 182.6 eV and 185.0 eV in the high-resolution spectrum of Zr (Figure 4b, Table 2). The two peaks corresponding to the 3d_5/2_ and 3d_3/2_ of Zr [59,60]. 

### 3.2. Hydrogenation of Furfuryl Alcohol

The Pd/UiO-66-v showed high activity in the hydrogenation of FA to THFA. The conversion was very low (<1%) when the reaction was conducted without any catalysts (Table 3, Entry 1). When the UiO-66-v without the introduction of Pd was used as catalysts, the conversion was also very low (2%) (Table 3, Entry 2). This showed that the Pd was responded for the conversion of FA. The hydrogenation reaction was then performed in 0.5 MPa H_2_ at 303 K in water in the presence of Pd/UiO-66-v. After reacted for 2 h, 28% of FA was converted with 92% selectivity to THFA (Table 3, Entry 3). As the H_2_ pressure increased from 0.5 MPa to 4 MPa, the conversion of FA increased from 28% to 92% while the selectivity to THFA kept around 90% (Table 3, Entries 3–7). On the other hand, when the reaction time was prolonged to 12 h at 0.5 MPa H_2_, the FA was converted almost completely with 90% selectivity to THFA (Table 3, Entry 8). The results showed the Pd/UiO-66-v was very effective for the hydrogenation of FA to THFA, even at mild conditions in water. The selectivity kept stand as the reaction time prolonged was also proved by the reaction progress profiles reacted at 4 MPa H_2_ (Appendix A). As the reaction time increased from 0.5 h to 2.5 h, the conversion of FA increased from 27% to 99%. The selectivity to THFA kept around 90% as the increasing reaction time. The effect of the solvent on the hydrogenation of FA was studied. When alcohols were used as the solvent, the selectivity to THFA maintained while the conversion decreased a lot. This decrease in activity should be interpreted from the perspective of the interaction between solvent and FA based on our recent work [57]. The FA had stronger interaction with alcohols than with water. When the alcohols were used as solvents, more energy was needed for the FA to get off the solvent to the surface of catalyst than in water solvent. This difference in energy demands should be responding for the different activity in alcohols or water, especially at a lower temperature. The water was the most suitable solvent for the hydrogenation of FA on Pd/UiO-66-v.

The effect of substrate concentration on the hydrogenation of FA was investigated by changing the concentration of FA with the same catalyst loading. When the reaction time was restricted at 2 h, the TOF for the hydrogenation of FA first increased from 202 h^−1^ to 286 h^−1^ when the concentration increased from 0.5 mol L^−1^ to 1.0 mol L^−1^ (Table 4, Entries 1,2). Then, the TOF decreased as the concentration continued to rise (Table 4). So, the concentration of 1.0 mol·L^−1^ should be the suitable concentration for this reaction when using Pd/UiO-66-v as the catalyst in water. However, a TOF as high as 187 h^−1^ was obtained at a concentration of 2.5 mol L^−1^. This showed that the catalyst could work at a higher concentration of FA. To test whether a higher conversion could be obtained at a high concentration of FA, the reaction time was prolonged for a different concentration of FA. 96% conversion of FA with 86% selectivity to THFA was acquired after 4 h when the concentration of FA was 1.0 mol·L^−1^ (Table 4, Entry 3). The TOF was 211 h^−1^ which was lower than only reacted for 2 h. A similar decrease in TOF was also observed at higher concentration of FA. 88%, 73%, and 58% conversion of FA were respectively obtained when the concentration of FA increased from 1.5 mol·L^−1^ to 2.5 mol·L^−1^ (Table 4, Entries 5, 7, and 9). The TOF was 193 h^−1^, 161 h^−1^, and 128 h^−1,^ respectively, which was lower than the value obtained at 2 h. The TOF should relate to the concentration of FA. As the time prolonged, the concentration of FA decreased, which led to the decrease in TOF. The selectivity to THFA was between 83% and 91%, showing the high selectivity to the hydrogenation furan rings. 

The Pd/UiO-66-v was used to hydrogenate different furan compounds to investigate the ability for hydrogenation of furan ring. As the 2-methylfuran, 2,5-dimethylfuran, and 5-methylfuran-2-methanol were insoluble in water, the hydrogenation reaction was conducted in ethanol. The conversion of 2-methylfuran was 99% after 2.5 h, even in ethanol. The selectivity to 2-methyltetrahydrofuran was 90% (Table 5, Entry 2). When 5-methylfuran-2-methanol, 2,5-dimethylfuran, and 2,5-di(hydroxymethyl)furan were used as the substrate, the compounds could not be transformed completely in 2.5 h (Table 5, Entries 3–5). The main product was the compounds of hydrogenation of the furan ring. To get high conversion of 5-methylfuran-2-methanol, 2,5-dimethylfuran, and 2,5-di(hydroxymethyl)furan, the reaction time was prolonged to 24 h. The conversion for the three furan derivatives was over 95% with the main product of the hydrogenation of furan ring (Table 5, Entries 3–5). Those results showed the Pd/UiO-66-v possessed high activity and selectivity for the hydrogenation of the furan ring.

The reuse performance of Pd/UiO-66-v was investigated by catalyst recycling experiments (Figure 5). The catalyst was recovered and washed by ethanol and water after the reaction and used for the next cycle. The selectivity to THFA kept around 90% in all the cycles, however, the conversion decreased to 72% in the third use. This decrease should be mainly ascribed to the loss of catalyst in the recovery process. We increased the reaction time to 3.5 h in the next cycle and the conversion increased to 98%, with 95% selectivity to THFA. The conversion was 89% with 92% selectivity to THFA in the fifth cycle after 4 h. 

## 4. Conclusions

In conclusion, UiO-66-v with vinyl in the benzene ring was successfully synthesized. The uniformly distributed vinyl was an effective reductant for the facile preparation of Pd/UiO-66-v with a narrow size distribution of Pd nanoparticles. The frame structure and high thermal stability was maintained after the introduction of Pd into UiO-66-v. The Pd/UiO-66-v showed high activity and selectivity for the hydrogenation of FA to THFA under mild conditions and could be used for the hydrogenation of furan ring in a series of furan derivatives. The Pd/UiO-66-v could be used for five times with no decrease in selectivity.

## Figures and Tables

**Figure 1 nanomaterials-09-01698-f001:**
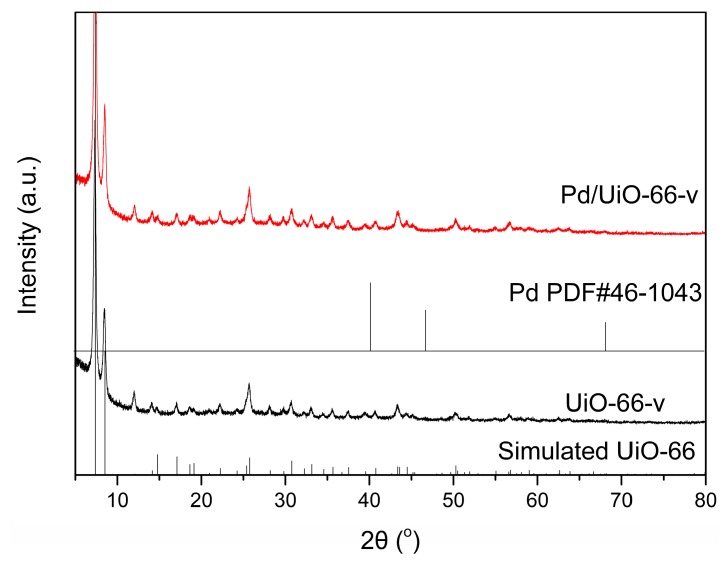
The XRD patterns of UiO-66-v, Pd/UiO-66-v, simulated UiO-66, and Pd PDF#46-1043.

**Figure 2 nanomaterials-09-01698-f002:**
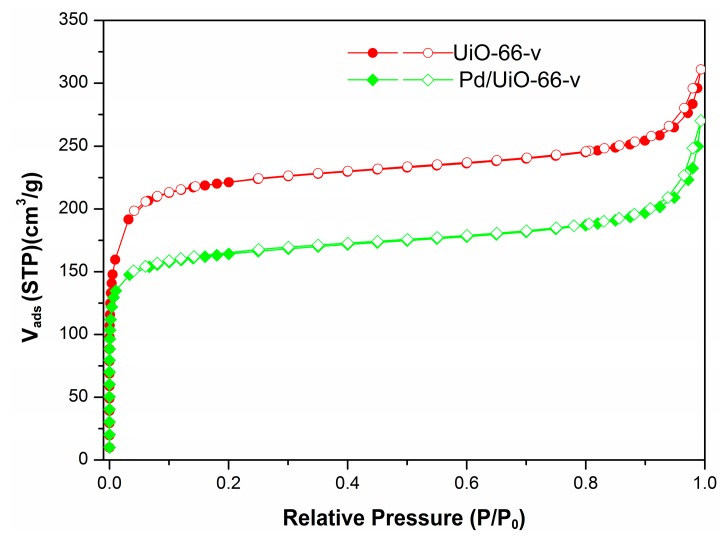
N_2_ adsorption/desorption isotherms of the UiO-66-v and Pd/UiO-66-v.

**Figure 3 nanomaterials-09-01698-f003:**
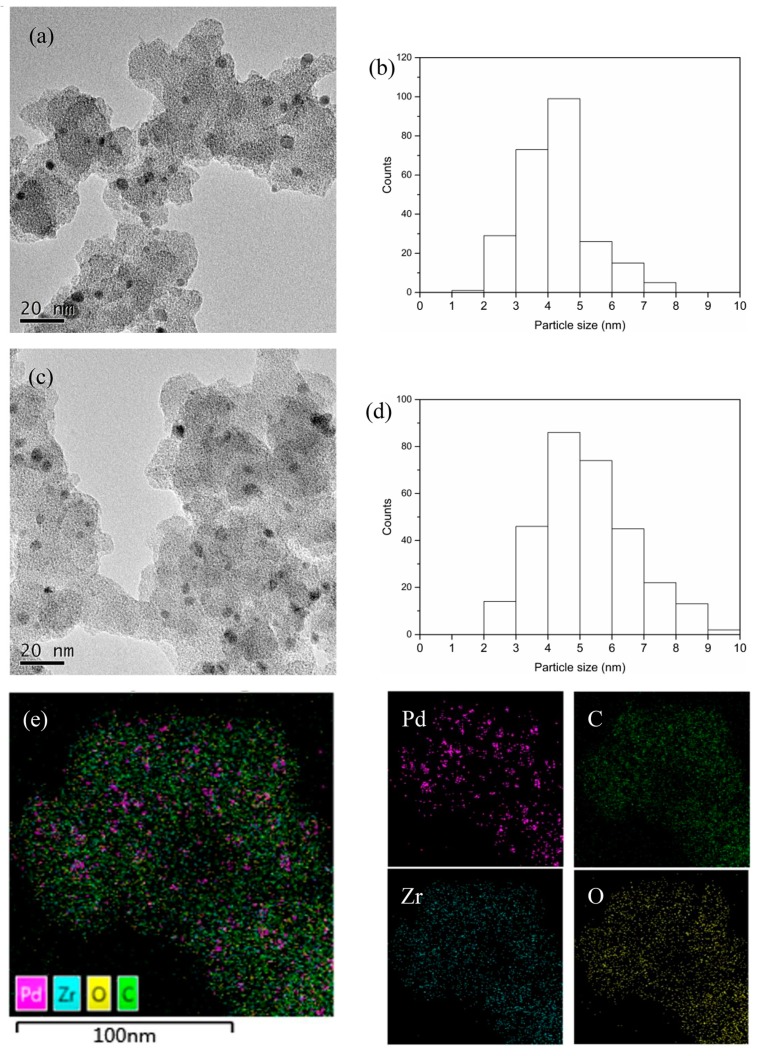
The TEM images and the corresponding particle size distribution of Pd/UiO-66-v (**a**,**b**) and the used catalyst (**c**,**d**) and the EDS mappings for Pd/UiO-66-v (**e**).

**Figure 4 nanomaterials-09-01698-f004:**
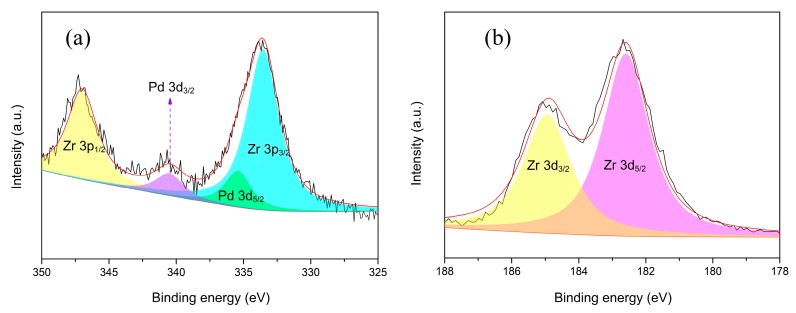
XPS spectra of Pd/UiO-66-v: high-resolution spectrum of Pd 3d (**a**) and Zr 3d (**b**).

**Figure 5 nanomaterials-09-01698-f005:**
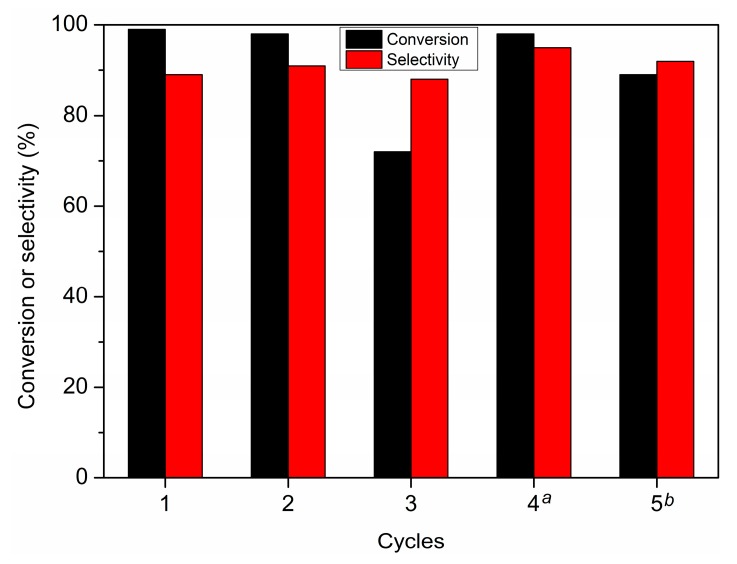
Cycle experiments for the hydrogenation of FA in water. *^a^* reaction time was 3.5 h. *^b^* reaction time was 4 h.

**Table 1 nanomaterials-09-01698-t001:** The structural properties of as-synthesized samples.

Entry	Sample	S_BET_ (m^2^·g^−1^)	S_t_-plot (m^2^·g^−1^)	PV ^1^ (cm^3^·g^−1^)	PV ^2^ (cm^3^·g^−1^)	Pd Content ^3^ (wt.%)
1	UiO-66-v	862	726	0.42	0.28	-
2	Pd/UiO-66-v	630	514	0.41	0.20	2.42 (2.35) ^4^

^1^ Total pore volume, ^2^ t-Plot micropore volume, ^3^ determined by the ICP-AES characterization, ^4^ Numbers in brackets referred to the Pd content for used catalyst.

**Table 2 nanomaterials-09-01698-t002:** The binding energy for Pd and Zr in Pd/UiO-66-v.

Element	Pd 3d_5/2_	Pd 3d_3/2_	Zr 3p_3/2_	Zr 3p_1/2_	Zr 3d_5/2_	Zr 3d_3/2_
B.E. (eV)	335.4	340.6	333.5	347.0	182.6	185.0

**Table 3 nanomaterials-09-01698-t003:** Hydrogenation of FA under different solvent, pressure, and reaction time ^1^.

Entry	Catalysts	Solvent	P (MPa)	Time (h)	Conversion (%)	Selectivity (%)	Carbon Balance
1	None	H_2_O	4	2	<1	n.d.	>99
2	UiO-66-v	H_2_O	4	2	2	n.d.	98
3	Pd/UiO-66-v	H_2_O	0.5	2	28	92	98
4	Pd/UiO-66-v	H_2_O	1	2	52	91	95
5	Pd/UiO-66-v	H_2_O	2	2	79	90	92
6	Pd/UiO-66-v	H_2_O	3	2	88	90	91
7	Pd/UiO-66-v	H_2_O	4	2	92	91	92
8	Pd/UiO-66-v	H_2_O	0.5	12	99	90	90
9	Pd/UiO-66-v	methanol	4	2	14	94	99
10	Pd/UiO-66-v	ethanol	4	2	38	93	97
11	Pd/UiO-66-v	isopropanol	4	2	2	96	>99

^1^ Reaction conditions: FA solution (1 mmol, 2 mL), 303 K, Pd/UiO-66-v (0.228 mol%, 10.0 mg) or UiO-66-v (10.0 mg), methyltetrahydrofuran and dimer (2,2′-(oxybis(methylene))difuran) were detected as by products, n.d. (not detected).

**Table 4 nanomaterials-09-01698-t004:** Effect of concentration on the hydrogenation of FA to THFA ^1^.

Entry	C (mol·L^−1^)	Time (h)	TOF (h^−1^)	Conversion (%)	Selectivity (%)	Carbon Balance
1	0.5	2	202	92	91	92
2	1.0	2	286	65	90	94
3	1.0	4	211	96	86	87
4	1.5	2	257	39	88	95
5	1.5	6	193	88	85	87
6	2.0	2	255	29	89	97
7	2.0	8	161	73	83	88
8	2.5	2	187	17	91	98
9	2.5	10	128	58	83	90

^1^ Reaction conditions: FA aqueous solution (2 mL), 4 MPa H_2_, 303 K, Pd/UiO-66-v (0.228 mol%, 10.0 mg), methyltetrahydrofuran and dimer (2,2′-(oxybis(methylene))difuran) were detected as by products. TOF: moles of FA converted/ (time × moles of Pd).

**Table 5 nanomaterials-09-01698-t005:** Hydrogenation of furan derivatives on Pd/UiO-66-v ^1^.

Entry	Substrate	Solvent	Conversion (%) ^2^	Selectivity (%) ^2^	Carbon Balance
1	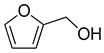	H_2_O	99	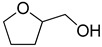	89	89
2	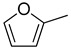	ethanol	99	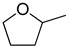	90	90
3	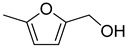	ethanol	44 (97)	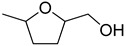	93 (91)	97 (91)
4	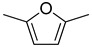	ethanol	11 (98)	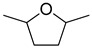	90 (90)	99 (90)
5	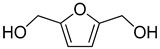	H_2_O	18 (99)	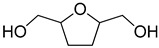	95 (92)	99 (92)

^1^ Reaction conditions: furan derivatives solution (2 mL), Pd/UiO-66-v (0.228 mol%, 10.0 mg), 303 K, 2 MPa H_2_, 2.5 h. ^2^ Numbers in brackets referred to the reaction time of 24 h.

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
