# Peer review of "Facile Preparation of Pd/UiO-66-v for the Conversion of Furfuryl Alcohol to Tetrahydrofurfuryl Alcohol under Mild Conditions in Water"

_nanomaterials, 2019, doi:10.3390/nano9121698_

Round 1
Reviewer 1 Report
The manuscript by Yanliang Yang et al. reports the preparation of zirconium UiO-66-type coordination polymer modified by palladium nanoparticles generated by in situ reduction of palladium(II) chloride by ethenyl substituents in MOF linkers. The material demonstrated good catalytic activity in furfuryl alcohol hydrogenation. The manuscript may be of interest to researchers involved in biomass conversion studies and MOF catalysis.
I have a few remarks that the authors could take into a consideration:
The authors found the BET surface area of their UiO-66-v to be about 860 m2g-1 and state that it is close to the value reported for unmodified UiO-66. However, that is not the case, since the BET surface area for UiO-66 is about 1600 m2g-1 (see e.g. Chem. Commun., 2013,49, 9449). In their catalytic studies section, the authors should present the results of blank experiments (without Pd and UiO-66-v and with unmodified UiO-66-v) to demonstrate the key role of Pd nanoparticles in the catalytic reduction. The bet selectivity of the reduction was found to be about 90%. If it is possible, other products should be listed somewhere in the manuscript to give a better understanding of the reaction and overall catalysis efficiency. In Table 2, I suggest to include the catalyst load in mole percent, not only in mg. Figures S10-15 with the mass spectra are not referenced anywhere in the text.
Author Response
The authors thank for the reviewer’s comments.
Please see the attachment.

Reviewer 2 Report
The authors present the work on furfuryl alcohol hydrogenation using Pd/UiO-66-v catalyst. The topic is of importance as the obtention of highly valued molecules from biomass is the base of biorefineries. In my opinion the work could be of interest of Nanomaterials journal but there are some controversial points that must be changed before acceptation. My remarks are given below:
The authors should clearly indicated why they choose the hydrogenation of FA and not furfural? Did tried to hydrogenate furfural directly to THFA? Experimental part should be improved. Especially the characterization part. Authors indicated only the commercial names of the equipments. The conditions of the analysis must be given. For example, XPS analysis the important information are missing: the pass energy? the deconvolution constraints? Shape line? Figure 2 should be presented differently. The diffraction peaks originated from the Pd should be presented in an insert. From TEM images the mean particle size is about 4-5 nm so they should be visible in XRD. If not the authors should commented on it. The XPS analysis must be improved. It is very difficult to check the binding energy of the peaks. It would be interesting to put the values in a Table. (Pd, Zr, O, C). The doublet separation is quite low (5.1 eV). The authors should explained. The major issue concerning XPS is the Pd peak deconvolution. For 3 d metals like Pd, the 5/2 peak of metallic Pd must be asymmetric. This should permit to put second component to the spectra (oxidized palladium). Catalytic activity results presented in Table 2 must contain the carbon balance values. Concerning the results it is quite strange to see low conversions when alcohols are using as a solvent. Several works with isopropanol have been already reported. Authors should give some hypothesis. The authors used TOFs but there is no explanation how the TOFs were calculated. This should be given in Experimental part. It is very difficult to calculate the exact number of active species. Perhaps it would be better to present TON instead of TOF. Water is green but is not a perfect solvent for the hydrogenation reactions. The leaching of metal can occur. Did authors checked the leaching after catalytic tests?
Author Response

(The authors gave the same response as above.)

Round 2
Reviewer 1 Report
The authors took great effort to improve the manuscript which can now be accepted for publication
Reviewer 2 Report
The authors present their revised article on Pd/IOU-66v catalysts for furfuryl alcohol hydrogenation. The manuscript was improved. The authors answered to all referees remarks and performed various changes in the original version of the manuscript. The revised version was improved in the introduction as well as in the presentation of the results. The carbon balance was added to the Tables as required. Discussion of the results was also improved. Taking all theses aspects into account I recommend this article for publication in the Nanomaterials journal in the present form.